# Efficacy and Safety of Lenvatinib in Hepatocellular Carcinoma Patients with Liver Transplantation: A Case-Control Study

**DOI:** 10.3390/cancers13184584

**Published:** 2021-09-12

**Authors:** Yen-Yang Chen, Chao-Long Chen, Chih-Che Lin, Chih-Chi Wang, Yueh-Wei Liu, Wei-Feng Li, Yen-Hao Chen

**Affiliations:** 1Department of Hematology-Oncology, Kaohsiung Chang Gung Memorial Hospital, Chang Gung University College of Medicine, Kaohsiung 833, Taiwan; chenyy@cgmh.org.tw; 2Division of General Surgery, Department of Surgery, Kaohsiung Chang Gung Memorial Hospital, Chang Gung University College of Medicine, Kaohsiung 833, Taiwan; clchen@cgmh.org.tw (C.-L.C.); chihchelin@cgmh.org.tw (C.-C.L.); chihchiwang@cgmh.org.tw (C.-C.W.); anthony0612@cgmh.org.tw (Y.-W.L.); webphone@cgmh.org.tw (W.-F.L.); 3Department of Nursing, Meiho University, Pingtung 912, Taiwan; 4School of Medicine, Chung Shan Medical University, Taichung 402, Taiwan

**Keywords:** lenvatinib, hepatocellular carcinoma, liver transplantation, survival, safety

## Abstract

**Simple Summary:**

Growing evidence has reported the role of sorafenib in hepatocellular carcinoma (HCC) patients with liver transplantation (LT). However, the clinical impact of lenvatinib in this population is limited. Our study enrolled 10 HCC patients who received lenvatinib after LT in our institute. Partial response was 20% and disease control rate was 70%. The median progression-free survival and overall survival were 3.7 and 16.4 months, respectively. Adverse events (AEs) were predominantly grade 1–2 in severity, and the majority of patients tolerated. Additionally, 25 HCC patients without LT who underwent lenvatinib treatment were identified as the control group; there was no significant difference in survival or AEs between these two groups. The significance of our study is that it is the first to investigate the efficacy and safety of lenvatinib among HCC patients with LT. It provides more information to physicians about the role of lenvatinib in this special population in clinical practice.

**Abstract:**

Tumor recurrence is the most common cause of death in hepatocellular carcinoma (HCC) patients who received liver transplantation (LT). Recently, lenvatinib was approved for the systemic treatment of unresectable HCC patients; however, the role of lenvatinib in HCC patients after LT remains unclear. There were 56 patients with recurrent HCC after LT from 2008 to 2018 in our institute, and 10 patients who received lenvatinib were identified. Additionally, to understand the difference in the clinical impact of lenvatinib in the LT and non-LT settings, 25 HCC patients without LT who underwent lenvatinib treatment were identified from our HCC database and regarded as the control group. In the LT group, partial response was 20% and stable disease was 50%, resulting in a disease control rate of 70%; the median progression-free survival (PFS), time to treatment failure (TTF) and overall survival (OS) were 3.7, 3.6 and 16.4 months, respectively. Adverse events (AEs) were predominantly grade 1–2 in severity, and the majority of patients tolerated the side effects. There was no significant difference in PFS/OS, and we observed a similar pattern of AEs between these two groups. Our study confirms the comparable efficacy and safety of lenvatinib in HCC patients with LT and non-LT in clinical practice.

## 1. Introduction

Hepatocellular carcinoma (HCC) is one of the most common aggressive malignancies worldwide and is the most common cause of death in patients with cirrhosis in Taiwan [1,2,3,4]. Liver transplantation (LT) offers a potentially curative therapeutic option for selected patients; the probability of HCC recurrence after LT is strongly associated with the pre-LT stage, leading to the development of restrictive criteria, such as the Milan criteria [5]. For patients who meet the Milan criteria, the outcome of HCC with LT is similar to the results obtained in those with non-tumor indications [6]. However, there is still an approximately 8–20% chance of HCC recurrence in these patients, leading to the most common cause of death in HCC patients who receive LT [7,8,9,10]. Surgical resection of intrahepatic or extrahepatic lesions is the gold-standard therapeutic choice with curative intent for amenable patients; other locoregional therapies, such as radiofrequency ablation or endovascular embolization, are alternative options [8,9]. Nevertheless, for patients with extrahepatic spread or who are not indicated for locoregional therapies, systemic treatment may be considered.

Sorafenib is a small, oral, multi-kinase inhibitor that targets the vascular endothelial growth factor (VEGF) receptors 2 and 3 (VEGFR-2/VEGFR-3), platelet-derived growth factor receptor (PDGFR) pathway, and mitogen-activated protein kinase (MAPK) pathway, leading to the inhibition of tumor cell proliferation and angiogenesis. Two phase III randomized control trials showed the safety and efficacy of sorafenib in advanced HCC patients who are not amenable to surgical resection or other locoregional therapies [11,12]. In these studies, sorafenib was proven to prolong overall survival; however, patients with LT were not enrolled and were ineligible for previous phase II/III trials [11,12,13]. Several retrospective cohort studies have reported that sorafenib has a heterogeneous treatment response and survival outcome in HCC patients with LT. The median duration of sorafenib ranged from 2.2 months to 10.2 months. Additionally, adverse events (AEs) included skin rash, hemifacial spasm, diarrhea, and fatigue; 92% of the patients experienced grade 3–4 diarrhea, and 77% discontinued sorafenib [14,15,16,17,18,19]. This may be related to many confounding factors: first, performance status after LT is an important issue, which depends on the complication of operation, side effects of immunosuppressants, and risk of opportunistic infections; second, previous studies have demonstrated that the period early after LT is associated with poor prognosis, indicating that a shorter recurrence time may reduce the effect of sorafenib; other prognostic variables were reported, such as the tumor location, serum alpha-fetoprotein (AFP) level, sites of distant metastasis, and number of organ involvement [8,20,21].

Lenvatinib, a newly developed tyrosine kinase inhibitor (TKI) that blocks VEGFR, PDGFR, and fibroblast growth factor receptor (FGFR), has been approved as a first-line systemic treatment for HCC, according to the REFLECT trial [22,23]. Compared to sorafenib, lenvatinib has a non-inferior overall survival (OS) and a significantly superior progression-free survival (PFS), time to progression, and objective response rate (ORR) [22]. In addition, the sequential use of sorafenib followed by lenvatinib is another issue in clinical practice. The SELECT trial, which focused on lenvatinib in patients with radioiodine-refractory thyroid cancer, showed no significant difference in survival between patients who experienced prior sorafenib treatment and those who did not [24]. We also published a cohort study to investigate the efficacy and safety of lenvatinib in patients with HCC who received sorafenib [25]. The ORR was 27.5%; the median PFS and OS were 3.3 months and 9.9 months, respectively, which was non-inferior to the second-line treatment of HCC, such as ramucirumab or regorafenib [26,27]. In addition, immunotherapy, such as pembrolizumab, nivolumab or atezolizumab plus bevacizumab, has been approved for treatment in HCC patients by several randomized clinical trials [28]. On the other hand, the early phase clinical trial has shown promising results of the combination of lenvatinib and pembrolizumab as a front-line treatment in advanced HCC patients [29]. However, to the best of our knowledge, information regarding the clinical impact of lenvatinib in HCC patients with LT is limited. The current study was designed to investigate the efficacy and safety of lenvatinib in HCC patients with LT, including ORR, PFS, OS, presentation of AEs, and survival comparison between HCC patients with and without LT.

## 2. Materials and Methods

### 2.1. Patient

We retrospectively reviewed patients with HCC who underwent LT at Kaohsiung Chang Gung Memorial Hospital between January 2008 and December 2018. First, patients with a history of second primary malignancy or concurrent hepatocholangiocarcinoma were excluded. There were 56 HCC patients with LT who had tumor recurrence; the diagnosis of HCC recurrence was either confirmed by pathology or according to the non-invasive criteria of the American Association for the Study of Liver Disease (AASLD) guidelines [30,31]. After that, different therapeutic modalities were performed for them, including surgical resection, transarterial embolization (TAE)/transarterial chemoembolization (TACE), chemotherapy, radiotherapy, and sorafenib. Subsequently, patients who were not amenable or failed locoregional treatment were treated with sorafenib, followed by lenvatinib after progression to sorafenib. Finally, 10 patients who received lenvatinib were identified.

In addition, to investigate the role of LT in the prognosis of HCC patients treated with lenvatinib, a control group was established. We retrospectively reviewed the HCC database at Kaohsiung Chang Gung Memorial Hospital between January 2008 and December 2020; the inclusion/exclusion criteria were the same as those in the LT group. Finally, 25 HCC patients without LT who received lenvatinib after sorafenib failure were included in the control group. A flowchart of identifying these HCC patients is shown in Figure 1.

### 2.2. Lenvatinib Treatment and Safety Assessment

Lenvatinib was started at a targeted dose of 10 mg once daily for each patient; all patients had a body weight ≥ 60 kg, whether they were part of the LT group or the control group [22,25]. Patients visited the outpatient clinic every 2–4 weeks for assessment of AEs, hematological and biochemical tests, AFP, vital signs, physical examination, and body weight. AEs were graded according to the National Cancer Institute Common Terminology Criteria for Adverse Events (CTCAE) version 4.0; all grades of each AE were recorded, and the worst grade was specified [32].

### 2.3. Staging and Efficacy Assessment

HCC staging was determined, according to the Barcelona Clinic Liver Cancer (BCLC) staging classification at the time of lenvatinib treatment initiation [33]. AFP level was measured before starting lenvatinib for each patient. Each patient had at least one measurable target lesion for evaluation of treatment response by enhanced computed tomography (CT) or magnetic resonance imaging (MRI) of the liver every 2–3 months after commencement of lenvatinib. The treatment response to lenvatinib was independently assessed by two radiologists blinded to any information about the patients’ clinical data, in accordance with the guidelines of the modified Response Evaluation Criteria in Solid Tumors (mRECIST) [34].

### 2.4. Statistical Analysis

Baseline characteristics of the patients were analyzed. For categorical variables, the chi-square test was performed to compare the characteristic distribution in the two identified groups. PFS was calculated from the date of lenvatinib treatment to disease progression or death from any cause; time to treatment failure (TTF) was the interval from the initiation of lenvatinib to discontinuation from any reason; OS was defined as the time from the initiation of lenvatinib to death or last living contact. Survival was calculated, using the Kaplan–Meier method. Relative dose intensity (RDI) was defined as the actual dose divided by the standard dose. Eight-week RDI (8W-RDI) means the cumulative dose of lenvatinib within the first 8 weeks divided by the standard dose [35]. Statistical significance was set at *p* < 0.05. Data analyses were performed using the SPSS 19 software (IBM, Armonk, NY, USA).

### 2.5. Ethics Statement

The current study was approved by the Chang Gung Medical Foundation Institutional Review Board (201900611B0). All methods were performed in accordance with the approved guidelines. Written informed consent was waived by the Chang Gung Medical Foundation Institutional Review Board, due to the retrospective design of this study.

## 3. Results

### 3.1. Patient Characteristics

Between January 2008 and December 2018, there were 528 HCC patients who received LT at Kaohsiung Chang Gung Memorial Hospital; 56 patients had tumor recurrence after LT. Surgical resection was performed in 20 patients. Additionally, eight patients received TAE/TACE, and 10 patients underwent chemotherapy. Furthermore, radiotherapy was administered to 4 patients, and sorafenib was prescribed for 14 patients. If locoregional treatment was not amenable, sorafenib was considered as a first-line systemic therapy; after sorafenib failure, lenvatinib was used as a second- or later-line of systemic treatment. Finally, 10 HCC patients with LT who received lenvatinib at our institution were identified. The characteristics of these HCC patients were documented at the time of lenvatinib administration. All patients were men with a mean age of 55.5 [range, 37–63] years. All of them were Child–Pugh classification A, whereas the BCLC staging classification was B in one (10%) patient and C in nine (90%) patients. In terms of viral hepatitis, six (60%) patients had hepatitis B virus (HBV) infection and four (40%) patients had hepatitis C virus (HCV) infection. The median duration from LT to HCC recurrence was 20.2 months. At recurrence, there were three (30%) patients with combined hepatic/extrahepatic spread, six (60%) patients with extrahepatic spread alone, and one (10%) patient with hepatic recurrence only. In other words, up to 90% of this cohort had extrahepatic spread as recurrence, including lung, bone, regional lymph nodes, and peritoneal seeding. At the time of analysis, the median follow-up period was 19.7 months for the 4 survivors and 13.6 months for all 10 patients. The demographic and clinical characteristics of the patients are presented in Table 1.

### 3.2. Lenvatinib Treatment and Efficacy

All patients in our study received sorafenib, whether as first-, second-, or later-line treatment; the median PFS of sorafenib was 4.6 months after which they received lenvatinib as a second- or later-line treatment. At the time of lenvatinib, neither main portal vein thrombosis nor macrovascular invasion was detected in these patients; 90% of this cohort had extrahepatic spread. According to the mRECIST criteria, no complete response was observed; partial response (PR) was noted in two (20%) patients, and stable disease (SD) in five (50%) patients, resulting in a disease control rate of 70%. The median PFS and OS were 3.7 months and 16.4 months, respectively.

### 3.3. Comparison of Prognosis between LT and Control Groups

In our study, a cohort of 25 HCC patients without LT who received lenvatinib as a second-line or later-line treatment was identified as the control group. In this group, the median follow-up period was 14.4 months for the 10 survivors and 10.5 months for all 25 patients. All patients had a body weight ≥ 60 kg. All characteristics were well-matched between the LT and control groups, except for HCV, macrovascular invasion, and hepatectomy before lenvatinib. Compared to the control group, the LT group had a higher percentage of HCV (40% versus 8%) and hepatectomy before lenvatinib (100% versus 60%); however, macrovascular invasion was more predominant in the control group than in the LT group (36% vs. 0%). The comparison of the baseline characteristics between the two groups is summarized in Table 2.

There was no significant difference in PFS (3.7 months vs. 4.2 months, Figure 2A) and TTF (3.6 months vs. 4.2 months, Figure 2B) between the LT group and the control group. OS analysis revealed no significant differences between these two groups, although a better OS was observed in the LT group compared to that in the control group (16.4 months vs. 12.0 months, Figure 2C). In addition, 8W-RDI was used to evaluate the correlation between the dose-intensity and clinical outcome. The cut-off value of 75% was based on a previous study [35]. Additionally, 8W-RDI ≥ 75% was mentioned in all patients in the LT group (100%) and 14 patients in the control group (56%). For all 35 patients in both groups, superior PFS was mentioned in patients with 8W-RDI ≥ 75%, compared to those with 8W-RDI < 75% (4.8 months vs. 3.2 months, *p* = 0.042, Figure 3A); better TTF was also found in 8W-RDI ≥ 75% group than 8W-RDI < 75% group (4.7 months vs. 3.2 months, *p* = 0.044, Figure 3B). Patients with 8W-RDI ≥ 75% were found to have superior OS in comparison with those with 8W-RDI < 75% (13.1 months vs. 6.2 months, *p* = 0.047, Figure 3C).

Eight (80%) patients in the LT group and 17 (68%) patients in the control group received post-lenvatinib anticancer treatment, including TAE/TACE, chemotherapy, radiotherapy, targeted therapy (such as cabozantinib, ramucirumab, and regorafenib), and immune checkpoint inhibitors. Of these patients, six (24%) patients received immune checkpoint inhibitors in the control group versus none (0%) in the LT group (Table 3).

### 3.4. Safety

All patients experienced AEs due to lenvatinib treatment. The most common AE in the LT group was hypertension (40%), followed by diarrhea (30%), fatigue (30%), palmar-plantar erythrodysesthesia (20%), and decreased appetite (20%). AEs were predominantly grade 1–2 in severity; grade 3 toxicities were rare, with only hypertension (10%) being manifested. No grade 4 or grade 5 (death) drug-related AEs occurred; treatment interruption or dose adjustment due to AEs was not mentioned. The majority of the patients tolerated the AEs of lenvatinib. None of the patients experienced graft rejection or deterioration of liver graft function. In addition, the patterns of AEs were similar between the LT and control groups. The incidence of drug-related AEs is shown in Table 4.

## 4. Discussion

Growing evidence has shown the role of lenvatinib in patients with HCC in real-world clinical practice; however, the information about lenvatinib in HCC patients with LT is very limited. Our study is the first to investigate the efficacy and safety of lenvatinib in this special population, including a similar response rate, PFS and OS compared to HCC patients without LT; in addition, there was no significant difference in adverse events between these two groups. Most patients received post-lenvatinib anticancer treatment; although immune checkpoint inhibitors were not given to HCC patients with LT, the OS was still not inferior to that in HCC patients without LT. The results of our study indicate the efficacy and safety of lenvatinib in HCC patients with LT in clinical practice.

In the past, the therapeutic modality for HCC patients not amenable to locoregional treatments was limited. In 2007, sorafenib was approved to significantly prolong overall survival in these patients and became the only recommended first-line systemic treatment for 10 years [11,12]. Recently, the REFLECT study, a phase III, open-label randomized controlled trial, showed non-inferior OS of lenvatinib in comparison with sorafenib [22]. Therefore, sorafenib and lenvatinib have both been regarded as first-line medications for TKI in patients with unresectable HCC. However, no prospective randomized controlled trial has focused on the efficacy and safety of sorafenib or lenvatinib; studies conducted for patients with recurrent HCC after LT remain scarce. To date, there have been several case series that demonstrated the effect and safety of sorafenib for recurrent HCC patients after LT [14,15,16,17,18,19,36,37,38]. The reported OS ranged from 5.4 months to 20.1 months; a highest disease control rate of 54% was mentioned in a Spanish study [36,37,38]. In addition, the majority of patients with recurrent HCC after LT could not tolerate sorafenib at a standard dose of 400 mg twice daily, resulting in frequent dose reduction. To date, there have been limited case reports regarding lenvatinib use in the post-LT setting. Pinero et al. reported that a non-cirrhotic male patient with chronic HCV had tumor recurrence 5 years after LT, and then received lenvatinib at a dose of 12 mg/day for at least 16 months [15]. Our study showed 10 HCC patients with tumor recurrence after LT, in which all of these patients received sorafenib followed by lenvatinib. To date, this is the largest cohort study of lenvatinib in the post-LT setting.

In our study, lenvatinib was prescribed after sorafenib and was regarded as a second- or later-line treatment. The reason for the sequential use of sorafenib–lenvatinib was related to the approval timeline, in which sorafenib and lenvatinib were approved in 2007 and 2008, respectively; therefore, sorafenib was the only one approved for systemic treatment at the time of tumor recurrence. To date, few studies have investigated the efficacy and safety of lenvatinib after sorafenib failure. Our previous study, which focused on the clinical impact of lenvatinib in HCC patients with sorafenib failure, showed that the median PFS and OS were 3.3 months and 9.9 months, respectively; the results revealed that lenvatinib was non-inferior to other second-line treatments for HCC, such as ramucirumab or regorafenib [26,27]. In order to explore whether the effect of lenvatinib is different between LT and non-transplantation settings, a cohort of HCC patients without LT was identified as the control group. There was no statistical difference in PFS and OS between the LT group and the control group, indicating the acceptable efficacy and safety of lenvatinib in recurrent HCC patients with LT.

According to the REFLECT trial, the initial dose of lenvatinib was 12 mg/day for patients with a body weight ≥ 60 kg; however, the mean lenvatinib dose intensity was 10.5 mg/day for these patients, corresponding to 88% of the planned starting dose in this group [22]. In addition, our previous studies also confirmed the efficacy and safety of lenvatinib at a dose of 10 mg/day, indicating that the ORR was 27.5%; the median PFS and OS were 3.3 months and 9.9 months, respectively. Furthermore, results revealed that lenvatinib is non-inferior to other second-line treatments for HCC, such as ramucirumab or regorafenib [26,27]. Therefore, all patients in our study, including the LT and control groups, had a body weight ≥ 60 kg and received lenvatinib at a dose of 10 mg once daily.

Although OS revealed no significant differences between these two groups, a better OS was found in the LT group, compared to that in the control group (16.4 months versus 12.0 months, respectively). The reason for better OS in the LT group may be summarized as follows: First, compared to the control group, patients in the LT group underwent liver transplantation, so they had better preserved liver function, contributing to a better response to lenvatinib. Second, the most recurrent pattern was extrahepatic spread; only 10% of patients had intrahepatic recurrence in the LT group, indicating a low tumor burden in the liver, resulting in better liver preservation and a lower risk of hepatic failure. Third, macrovascular invasion is a well-known poor prognostic factor; however, no macrovascular invasion was detected in the LT group, including the portal vein, hepatic vein, or inferior vena cava, contributing to a better OS. Fourth, there was a higher percentage of HCV in the LT group than in the control group. Bruix et al. reported that HCV is predictive of a superior OS benefit with sorafenib, but the mechanism is still unclear; it may be caused by the association of the inflammatory status and persistent viral replication [39]. Although there is still a lack of evidence regarding the impact of HCV in the lenvatinib setting, the mechanism by which HCV modulates the effect of sorafenib might be similar to that of lenvatinib.

Lenvatinib has been approved for thyroid cancer and HCC, but the initial dose is different for these two cancer types. For HCC, the dose of lenvatinib is adjusted based on the body weight, which is 12 mg/day for ≥ 60 kg and 8 mg/day for < 60 kg; however, the initial dose of lenvatinib is 24 mg/day for thyroid cancer patients. The rationale is poor liver preservation in HCC patients, leading to poor metabolism, poor tolerability, and higher toxicity. In our study, the profiles of AEs in the LT group were comparable to those in the control group. Almost all patients developed AEs, but most AEs were grade 1–2 in severity and no grade 4 or grade 5 (death) drug-related AEs were mentioned. Although immunosuppressive agents are routinely used in the LT group, they benefit from LT-induced better preserved liver function and lower tumor burden of the liver, contributing to better tolerability of lenvatinib.

The dose adjustment is an important issue of lenvatinib. First, the dose of lenvatinib in the treatment of HCC and thyroid cancer is different, and the reason may be associated with worse liver function in HCC patients, compared to thyroid cancer patients. Second, the recommended dose of lenvatinib in HCC is based on the actual body weight. However, the optimal dose of lenvatinib in HCC patients with LT is unclear. For HCC patients with LT, they have a new liver and better liver function, so the optimal dose of lenvatinib may be higher than the recommended dose; nevertheless, these patients need to receive an immunosuppressant to avoid acute or chronic rejection; this may also interfere with the effect of lenvatinib. Therefore, in order to answer the complex question, we designed this study, and our results demonstrated the similar efficacy and safety of lenvatinib between HCC patients with or without LT. In the future, a clinical trial focused on lenvatinib following liver transplantation in patients with HCC is warranted to validate the findings of our study. On the other hand, immunotherapy is still contra-indicated for HCC patients with LT at present, so the combination of lenvatinib with other targeted therapy or development of additional novel therapeutic agents may potentially augment the treatment options and improve overall survival of HCC patients with LT.

In our study, there was no deterioration of transplanted liver function with the combination of lenvatinib and immunosuppressant drugs; in addition, the efficacy and safety of lenvatinib in the LT group was similar to the control group, indicating the exclusion of the potentially negative prognostic role of immunosuppressant agents on lenvatinib.

There are several limitations to our study. First, the study had a small sample size of recurrent HCC patients with LT, so we could not identify the prognostic factors of PFS and OS. Second, the duration of the follow-up period may not be long enough, so the survival benefit may not be significant. Sorafenib and lenvatinib have both been approved for systemic treatment in HCC patients. There were several studies which enrolled many patients with HCC and focused on the efficacy and safety of these two drugs in real-world clinical practice. However, HCC patients with recurrence after LT is a special group with small patient numbers. Even though sorafenib has been approved and used for more than 10 years in clinical practice, there were only a few studies published in the past, and the enrolled patient numbers were around 9 patients to 31 patients [15]. This situation indicates the limitation of this systemic therapy in HCC patients with liver transplantation. Therefore, it is difficult to enroll many patients in one study to investigate the clinical impact of lenvatinib. However, to the best of our knowledge, this is the first cohort study designed to evaluate the efficacy and safety of lenvatinib in recurrent HCC patients with LT. Further studies on larger series or randomized controlled trials are needed to confirm the findings of this study.

## 5. Conclusions

Our study confirms the similar efficacy and safety of lenvatinib in HCC patients with LT and non-LT in clinical practice.

## Figures and Tables

**Figure 1 cancers-13-04584-f001:**
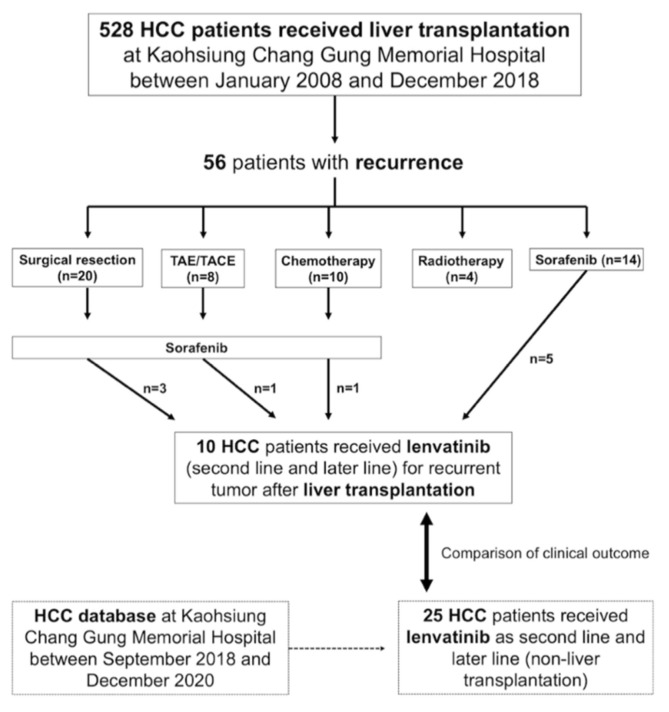
Flowchart for identifying the hepatocellular carcinoma patients who received lenvatinib for tumor recurrence after liver transplantation.

**Figure 2 cancers-13-04584-f002:**
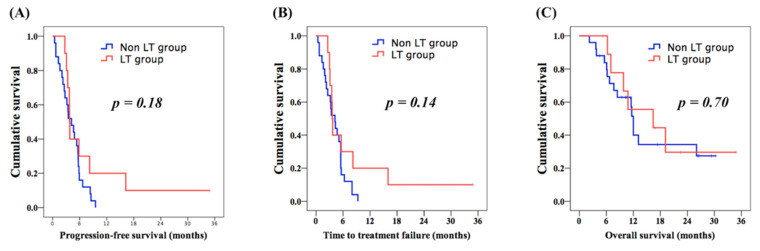
Comparison of Kaplan–Meier survival curves of progression-free survival (PFS), time to treatment failure (TTF) and overall survival (OS) between liver transplantation (LT) group and non-LT group. (**A**) PFS, (**B**) TTF and (**C**) OS.

**Figure 3 cancers-13-04584-f003:**
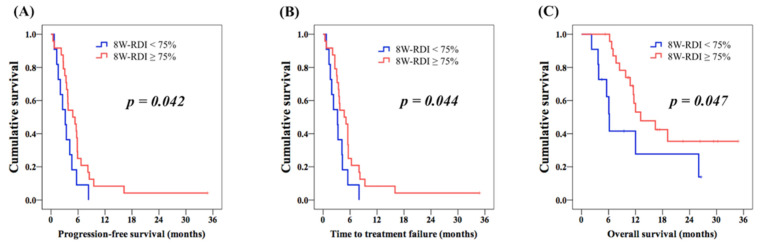
Kaplan–Meier survival curves comparing progression-free survival (PFS), time to treatment failure (TTF) and overall survival (OS) between patients with 8W-RDI ≥ 75% or 8W-RDI < 75%. (**A**) PFS, (**B**) TTF and (**C**) OS.

**Table 1 cancers-13-04584-t001:** Characteristics of 10 patients with recurrent unresectable hepatocellular carcinoma after liver transplantation.

Patient Number	Age/Sex	Body Weight (kg)	ECOG PS	Child-Pugh Classification	BCLC Classification	Viral Hepatitis Status	Immunosuppressant Agents	Duration from LT to Tumor Recurrence (Months)	Recurrent Status	Extrahepatic Spread at the Time of Recurrence	Metastatic Site at the Time of Recurrence	Duration of Prior Sorafenib (Months)	Best Response to Sorafenib	Main PVT at the Time of Lenvatinib	Macrovascular Invasion at the Time of Lenvatinib	Extrahepatic Spread at the Time of Lenvatinib	Metastatic Site at the Time of Lenvatinib	LN Involvement at the Time of Lenvatinib	AFP at the Time of Lenvatinib	Best Response to Lenvatinib
1	39/M	67.0	0	A	C	hepatitis B	Everolimus Tacrolimus Prolonged-release	22.0	Hepatic combined with extrahepatic	Yes	Regional lymph nodes	4.6	PR	No	No	Yes	Regional lymph nodes	Yes	<2.0	PR
2	37/M	65.7	1	A	C	hepatitis B	Sirolimus Tacrolimus (FK506)	49.2	Extrahepatic	Yes	lung	3.0	SD	No	No	Yes	Lung and regional lymph nodes	Yes	3.8	SD
3	54/M	66.6	1	A	C	hepatitis B	Tacrolimus (FK506) Sirolimus	10.2	Hepatic combined with extrahepatic	Yes	lung	30.7	PR	No	No	Yes	Lung	No	78.2	SD
4	62/M	78.2	1	A	C	hepatitis B	Tacrolimus (FK506)	18.3	Extrahepatic	Yes	bone	3.1	SD	No	No	Yes	Bone	No	<3.0	SD
5	63/M	61.0	1	A	C	hepatitis C	Everolimus Tacrolimus Prolonged-release	23.4	Extrahepatic	Yes	Peritoneal seeding	2.5	SD	No	No	Yes	Lung and peritoneal seeding	No	196.1	SD
6	39/M	67.6	0	A	C	hepatitis B	Tacrolimus (FK506) Sirolimus	14.9	Hepatic combined with extrahepatic	Yes	lung	5.1	PD	No	No	Yes	Lung	No	830.4	PD
7	58/M	73.1	0	A	C	hepatitis C	Sirolimus Mycophenolate mofetil	54.2	Extrahepatic	Yes	bone, lung	18.9	PR	No	No	Yes	Lung and bone	No	2942.0	PR
8	49/M	66.9	0	A	C	hepatitis B	Everolimus Tacrolimus (FK506)	34.0	Extrahepatic	Yes	lung	12.9	SD	No	No	Yes	Lung	No	2.7	SD
9	57/M	63.5	1	A	B	hepatitis C	Tacrolimus (FK506) Sirolimus	13.8	Hepatic	No	–	4.7	PD	No	No	No	–	No	57.9	PD
10	57/M	71.0	1	A	C	hepatitis C	Tacrolimus (FK506) Mycophenolate mofetil	13.0	Extrahepatic	Yes	bone	4.2	PD	No	No	Yes	Bone	No	46,142.3	PD

ECOG PS: Eastern Cooperative Oncology Group performance status; BCLC: Barcelona Clinic Liver Cancer; PVT: portal vein thrombosis; LT: liver transplantation; LN: lymph node; AFP: alpha-fetoprotein; PR: partial response; SD: stable disease; PD: progressive disease.

**Table 2 cancers-13-04584-t002:** Clinicopathological parameters in 35 patients with unresectable hepatocellular carcinoma who received lenvatinib.

Characteristics	Liver Transplantation Group ^#^ (*n* = 10)	Control Group (*n* = 25)	*p* Value
Age			
<60 years	8 (80.0%)	11 (44.0%)	0.05
≥60 years	2 (20.0%)	14 (56.0%)	
ECOG performance status			
0	3 (30.0%)	4 (16.0%)	0.35
1	7 (70.0%)	21 (84.0%)	
Sex			
Male	9 (90.0%)	22 (88.0%)	0.87
Female	1 (10.0%)	3 (12.0%)	
Child–Pugh classification			
A	10 (100.0%)	25 (100.0%)	1.0
BCLC staging classification			
B	2 (20.0%)	1 (4.0%)	0.13
C	8 (80.0%)	24 (96.0%)	
Hepatitis B			
Yes	6 (60.0%)	17 (68.0%)	0.65
No	4 (40.0%)	8 (32.0%)	
Hepatitis C			
Yes	4 (40.0%)	2 (8.0%)	0.023 *
No	6 (60.0%)	23 (92.0%)	
Macrovascular invasion (including main PVT) at the time of lenvatinib			
Yes	0 (0%)	9 (36.0%)	0.042 *
No	10 (100.0%)	16 (64.0%)	
Hepatectomy before lenvatinib treatment			
Yes	10 (100.0%)	15 (60.0%)	0.018 *
No	0 (0%)	10 (40.0%)	
Extrahepatic spread at the time of lenvatinib			
Yes	9 (90.0%)	20 (80.0%)	0.48
No	1 (10.0%)	5 (20.0%)	
AFP level > 400 at the time of lenvatinib			
Yes	2 (20.0%)	13 (52.0%)	0.08
No	8 (80.0%)	12 (48.0%)	

ECOG: Eastern Cooperative Oncology Group; BCLC: Barcelona Clinic Liver Cancer; PVT: portal vein thrombosis. ^#^ Lenvatinib used for tumor recurrence after liver transplantation * Statistically significant.

**Table 3 cancers-13-04584-t003:** Post-lenvatinib anti-cancer therapy in the 35 patients with unresectable hepatocellular carcinoma.

Category	Liver Transplantation Group ^#^ (*n* = 10)	Control Group (*n* = 25)
Any post-lenvatinib anti-cancer treatment	8 (80%)	17 (68%)
TACE/TAE	3 (30%)	3 (12%)
Chemotherapy	4 (40%)	7 (28%)
Radiotherapy	1 (10%)	1 (4%)
Clinical trials	0	2 (8%)
Palliative metastasectomy	2 (20%)	0
Target Therapy	4 (40%)	7 (28%)
Carbozantinib	2 (20%)	0
Regorafenib	2 (20%)	2 (8%)
Ramucirumab	0	1 (4%)
Sorafenib	0	1 (4%)
Thalidomide	1 (10%)	4 (16%)
Immune checkpoint inhibitors	0	6 (24%)
Atezolizumab plus bevacizumab	0	1 (4%)
Atezolizumab	0	1 (4%)
Nivolumab	0	3 (12%)
Pembrolizumab	0	1 (4%)

^#^ Lenvatinib used for tumor recurrence after liver transplantation; TAE: transarterial embolization; TACE: transarterial chemoembolization.

**Table 4 cancers-13-04584-t004:** Lenvatinib-related adverse events in the 35 patients with unresectable hepatocellular carcinoma.

Adverse Event	Liver Transplantation Group ^#^ (*n* = 10)		Control Group (*n* = 25)		*p* Value
	Any Grades	Grade 3	Any Grades	Grade 3	
Hypertension	4 (40.0%)	1 (10.0%)	12 (48.0%)	3 (12.0%)	0.98
Diarrhea	3 (30.0%)	0 (0%)	7 (28.0%)	1 (4%)	0.78
Decreased appetite	2 (20.0%)	0 (0%)	4 (16.0%)	0 (0%)	0.78
Decreased body weight	1 (10.0%)	0 (0%)	3 (12.0%)	0 (0%)	0.87
Fatigue	3 (30.0%)	0 (0%)	8 (32.0%)	1 (4.0%)	0.91
Palmar-plantar erythrodysesthesia	2 (20.0%)	0 (0%)	6 (24.0%)	0 (0%)	0.80
Nausea	1 (10.0%)	0 (0%)	3 (12.0%)	0 (0%)	0.87
Vomiting	1 (10.0%)	0 (0%)	2 (8.0%)	0 (0%)	0.85
Skin rash	1 (10.0%)	0 (0%)	3 (12.0%)	0 (0%)	0.87

^#^ Lenvatinib used for tumor recurrence after liver transplantation.

## Data Availability

Data are contained within the article.

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
