# Peer review of "Efficacy and Safety of Lenvatinib in Hepatocellular Carcinoma Patients with Liver Transplantation: A Case-Control Study"

_cancers, 2021, doi:10.3390/cancers13184584_

Round 1

Reviewer 1 Report

Dear Editor, thank you so much for inviting me to revise this manuscript about lenvatinib in HCC

This study addresses a current topic.

The manuscript is quite well written and organized. English could be improved.

Figures and tables are comprehensive and clear.

The introduction explains in a clear and coherent manner the background of this study.

We suggest the following modifications:

  • Introduction section: although the authors correctly included important papers in this setting, we believe a couple of studies should be cited within the introduction (PMID: 30993651 ; PMID: 34167433), only for a matter of consistency. We think it might be useful to introduce the topic of this interesting study.
  • Methods and Statistical Analysis: nothing to add.
  • Discussion section: Very interesting and timely discussion. Of note, the authors should expand the Discussion section, including a more personal perspective to reflect on. For example, they could answer the following questions – in order to facilitate the understanding of this complex topic to readers: what potential does this study hold? What are the knowledge gaps and how do researchers tackle them? How do you see this area unfolding in the next 5 years? We think it would be extremely interesting for the readers.

However, we think the authors should be acknowledged for their work. In fact, they correctly addressed an important topic in HCC, the methods sound good and their discussion is well balanced.

One additional little flaw: the authors could better explain the limitations of their work, in the last part of the Discussion.

We believe this article is suitable for publication in the journal although major revisions are needed. The main strengths of this paper are that it addresses an interesting and very timely question and provides a clear answer, with some limitations.

We suggest a linguistic revision and the addition of some references for a matter of consistency. Moreover, the authors should better clarify some points.

Reviewer 2 Report

The authors summarize about cases receiving lenvatinib for the HCC recurrent patients after liver transplantation. The lenvatinib treatment for the patients with liver transplantation is limited, and I think it is a valuable report. However, I ask you for discussion about the following points.

  1. How did you select the 25 patients for the comparison? Did you extract by using statistical method, such as propensity score? In the first place, there are few implication of the comparison. I regard that you had better report as case control study.
  2. Therefore, I want you to investigate clinical courses a little more deeply, such as relative dose intensity of lenvatinb, and time-to treatment failure. Wasn’t there the problem of the blood concentration of the immunoregulator?

Round 2

Reviewer 1 Report

The authors modified the paper according to our suggestions.

We recommend Acceptance in its current form.

Reviewer 2 Report

Thank you very much for giving opportunity to review this revised manuscript.

I agree to revised contents totally and regard that this is suitable for publication.